# Role of FXR in Renal Physiology and Kidney Diseases

**DOI:** 10.3390/ijms24032408

**Published:** 2023-01-26

**Authors:** Yanlin Guo, Guixiang Xie, Xiaoyan Zhang

**Affiliations:** Health Science Center, East China Normal University, Shanghai 200241, China

**Keywords:** Farnesoid X receptor, renal physiology, acute kidney disease, chronic kidney disease

## Abstract

Farnesoid X receptor, also known as the bile acid receptor, belongs to the nuclear receptor (NR) superfamily of ligand-regulated transcription factors, which performs its functions by regulating the transcription of target genes. FXR is highly expressed in the liver, small intestine, kidney and adrenal gland, maintaining homeostasis of bile acid, glucose and lipids by regulating a diverse array of target genes. It also participates in several pathophysiological processes, such as inflammation, immune responses and fibrosis. The kidney is a key organ that manages water and solute homeostasis for the whole body, and kidney injury or dysfunction is associated with high morbidity and mortality. In the kidney, FXR plays an important role in renal water reabsorption and is thought to perform protective functions in acute kidney disease and chronic kidney disease, especially diabetic kidney disease. In this review, we summarize the recent advances in the understanding of the physiological and pathophysiological function of FXR in the kidney.

## 1. Introduction

Nuclear receptors are a superfamily of ligand-dependent transcription factors, with typical domain structures and conserved sequences. There are 48 members of the human nuclear receptor superfamily which perform crucial roles in a variety of processes of physiology and pathophysiology, such as homeostasis, reproduction, development, metabolism, obesity, diabetes, hypertension and cancer. Therefore, they always attract attention from basic scientists, clinicians, and the pharmaceutical industry, and account for 3% of all human drug targets so far [1].

Nuclear receptor farnesoid X receptor (FXR) was first isolated from a rat liver cDNA library and named after its weak activation via supraphysiological farnesol, an intermediate in the mevalonate biosynthetic pathway, in 1995 [2,3]. Subsequently, bile acids were identified as endogenous FXR ligands and were given another name: bile acid receptors. As a bile acid sensor, FXR is a key modulator of enterohepatic circulation of bile acids, controlling transcription of key regulatory genes in bile acid synthesis, biliary bile acid secretion and trans-intestinal bile acid transport to the liver via portal blood circulation. FXR also serves as a metabolic regulator of glucose, lipid and energy metabolism. Because of its master functions in bile acid and lipid metabolism, FXR is considered a promising drug target for the therapy of bile-acid-related liver diseases. In 2016, obeticholic acid (OCA), a highly selective agonist of FXR, was approved by the U.S. Food and Drug Administration for treatment of primary biliary cholangitis (PBC), a chronic cholestatic liver disease. Currently, FXR agonists are tested in clinical trials for treatment of metabolic and gastrointestinal diseases, including type 2 diabetes (T2DM), metabolic syndrome and Non-Alcoholic Steatohepatitis (NASH).

The kidney is a key organ, managing water and solute homeostasis for the whole body, and kidney injury or dysfunction is associated with high morbidity and mortality. FXR exhibits high expression in the kidney, but little is known about its role in the kidney. Recently, we and other groups reported that FXR participates in renal water reabsorption and is involved in kidney diseases, including acute kidney injury and chronic kidney disease. This review summarizes the structure and expression of FXR, with a particular focus on the recent advancement of FXR in regulation of renal physiology and kidney diseases. 

## 2. The Structure and Expression of FXR

Two FXR protein families have been identified in animals. Derived from the nuclear receptor subfamily 1, group H, member *(NR1H) 4* and *NR1H5* genes, they are named FXRα and FXRβ, respectively. From fish to humans, FXRα is evolutionarily conserved among species. Due to the activation of two different promoters and the use of alternative splicing, the FXRα gene encodes four isoforms (FXRα1-α4) [4,5]. A recent study identified four novel splice variants (FXRα5–8) in human hepatocytes, which resulted from previously undetected exon skipping events. The possible roles of these novel isoforms in human liver require further investigation [6]. However, the FXRβ coding gene is a pseudogene in humans and primates [7]. FXR is highly expressed in the liver, intestine, kidney, and adrenal gland, with low levels of expression in adipose tissue and heart. The expression of FXRα1-α4 in different tissues remains unclear. Using Southern blot analysis, Zhang et al. found that mouse FXRα1-α4 had the highest expression in liver, and the expression levels of each isoform are similar. In addition, FXRα1 and FXRα2 are moderately expressed in the ileum and adrenal gland; FXRα3 and FXRα4 are highly expressed in the ileum and moderately expressed in the kidney [8]. More efforts are needed to analyze the tissue distribution of different FXR isoforms in the future to help elucidate their functions (Table 1).

FXR has a typical domain structure of a nuclear receptor (NR), including an N-terminal domain (NTD), DNA-binding domain (DBD), hinge region and ligand-binding domain (LBD). The N-terminal domain (NTD) contains a ligand-independent transcriptional activation domain (AF1) which can interact with other coregulator proteins such as peroxisome proliferator-activated receptor gamma (PPARγ) coactivator 1-alpha (PGC1α), E1A-binding protein p300 (P300) and nuclear receptor co-repressor 2 (NCOR2/SMRT) [9], and many post-translational modifications (PTM) including phosphorylation, SUMOylation, GlcNAcylation, acetylation and methylation can occur in this region [10]. PGC-1α functions as a transcriptional coactivator of FXR that directly interacts with FXR and enhances its transactivation activity [11]. Phosphorylation of FXR by protein kinase C promotes its transcriptional activity [12]. FXR O-GlcNAcylation at S62 results in increased FXR activity [13]. Some coactivators have intrinsic enzymatic activity and can modify NRs themselves. Acetylation of FXR by P300 increases its stability but reduces FXR-RXRα heterodimerization, leading to a reduction in FXR target gene expression [14]. The zinc finger DNA-binding domain (DBD) is the most conserved domain, containing two zinc finger motifs that recognize specific DNA sequences [15]. The flexible hinge region is a linker between the DBD and the LBD, which is also a site for the regulation of PTMs such as the NTD [10]. Ligand-binding domain (LBD) is composed of 11 α-helices and four β-strands that fold into three parallel layers to form an alpha helical sandwich. This fold forms a hydrophobic C-terminal ligand binding pocket (LBP) and AF-2. LBD can bind to ligands and AF-2 can recruit co-activators [16,17,18]. When ligands are present, FXR binds to specific DNA sequences in the promoter of target genes in the form of a monomer or a heterodimer with retinoid X receptor (RXR), the common partner for NRs, to regulate gene transcription [19].

## 3. The Ligands of FXR

Bile acids are important endogenous agonists of FXR. Cholic acid (CA) and chenodeoxycholic acid (CDCA) are two primary bile acids synthesized in the liver. The secondary bile acids lithocholic acid (LCA) and deoxycholic acid (DCA) are generated from CA and CDCA. The potency of bile acids in activating FXR is ranked as: CDCA > DCA > LCA > CA [20]. OCA is a semi-synthetic derivative of CDCA, also known as 6-ethyl-CDCA and INT-747, which is the first FXR agonist to enter clinical research [21]. However, OCA has side effects such as dose-dependent pruritis, which can lead to treatment discontinuation in ~1–10% of patients. EDP-305 is another steroid FXR agonist studied for the treatment of NASH and PBC [22]. In order to reduce side effects and improve the therapeutic effect, non-steroidal FXR agonists have been gradually synthesized. These include GW4064 [23], Cilofexor [24], Tropifexor [25], Nidufexor [26] and others [27,28,29,30,31,32,33,34]. Therefore, FXR agonists are actively used to treat a variety of metabolic diseases in the clinics [35] (Table 2).

Some bile acids are considered FXR Antagonists. Tauro-β-muricholic acid (T-β-MCA), an endogenous FXR antagonist, was reported to inhibit activation of FXR [36]. It is reported that glycine-β-muricholic acid (Gly-MCA) inhibited FXR signaling exclusively in the intestine, resulting in a decrease in the serum and intestine ceramide level and an improvement in metabolic dysfunction in obese mice [37]. Ursodeoxycholic acid (UDCA) exerts FXR-antagonistic effects on bile acid and lipid metabolism in morbid obesity, although it is a commonly used therapeutic agent in cholestatic liver disease [38]. More recently, Brevini et al. found that UDCA reduces angiotensin-converting enzyme 2 (ACE2) expression by inhibiting FXR activity, resulting in reduced susceptibility to SARS-CoV-2 infection [39]. Therefore, FXR antagonists are also actively used to treat multiple diseases in the clinics (Table 3).

## 4. The General Function of FXR

### 4.1. FXR and Bile Acid Metabolism

FXR plays critical roles in bile acid homeostasis [46]. FXR inhibits bile acid synthesis: in the liver, FXR decreases the expression of cytochrome P450 (CYP)7A1 and CYP8A1, which are rate-limiting enzymes in bile acid biosynthesis of cholesterol [47,48]. In the intestinal enterocytes, FXR induces the expression of fibrotic growth factor 15 (FGF15; FGF19, the orthologue humans of FGF15), which travels through the portal vein to the liver and activates the FGF receptor 4 (FGFR4)/β-Klotho complex, thereby inhibiting transcription of CYP7A1 and CYP8A1 [49]. FXR reduces bile acid accumulation in hepatocytes and enterocytes: in liver, FXR can inhibit the expression of Na^+^-taurocholate cotransporting polypeptide (NTCP) and organic-anion-transporting polypeptides (OATP) at the sinusoidal membrane to reduce hepatocytes’ reabsorption of bile acids in the portal vein [50]. FXR promotes the excretion of bile acids into bile by activating transporters on the apical membrane surface, including multidrug-resistance-associated protein (MRP)2/3 [51], bile salt export pump (BSEP) [52] and multidrug resistance (MDR)2/3 [53]. FXR also promotes bile acid efflux into blood circulation by inducing expression of MRP4 and organic solute transporter (OST)α/OSTβ in the basolateral membrane [54]. In the intestine, FXR inhibits apical sodium-dependent bile acid transporter (ASBT) at the apical membrane surface [55], thereby reducing the reabsorption of bile acids in the intestinal epithelium, upregulating ileal bile acid-binding protein (IBABP) and promoting its movement from the apical membrane to the basolateral membrane [56]. Moreover, FXR promotes the transportation of bile acids to the portal vein and their subsequent return to the liver via upregulation of the expression of OSTα and OSTβ on the basolateral membrane surface of the enterocyte [57].

### 4.2. FXR and Glucose Metabolism

FXR plays diverse roles in glucose metabolism. The activation of FXR increases glycogen synthesis by inhibiting glycogen synthase kinase-3 beta (GSK3β) gene expression, which phosphorylates and subsequently inactivates glycogen synthase [58,59], reduces glycolysis by suppressing the transcriptional activity of ChREBP [60] and decreases the expression of multiple gluconeogenic genes, including phosphoenolpyruvate carboxykinase (PEPCK) and glucose 6-phosphatase (G-6-Pase) [58,61], resulting in a decrease in gluconeogenesis and serum glucose. However, several studies found that FXR antagonist exhibited a beneficial effect on glucose metabolism in T2DM mice, although the exact mechanism remains unclear [62]. Furthermore, FXR increases glucose-stimulated insulin secretion in islets [63] by inducing the expression of glucose-regulated transcription factor Krueppel-like factor 11 (KLF11) [64], adenylyl cyclase 8 (ADCY8) [65] and the transient receptor potential ankyrin 1 (TRPA1) channel [66].

### 4.3. FXR and Lipid Metabolism

FXR reduces the plasma low-density lipoprotein cholesterol (LDL-C) level [67] by inducing the internalization and degradation of the LDL particle [68]. Moreover, it was found that FXR knockout mice displayed elevated plasma high-density lipoprotein cholesterol (HDL-C) due to the reduced expression of reverse cholesterol transport gene, scavenger receptor class B member 1 (SCARB1) and ATP-binding cassette (ABC) transporters G5 (ABCG5) and G8 (ABCG8), which facilitated the removal of HDL-C from the blood [69,70]. Finally, FXR can also reduce the accumulation of cholesterol in hepatocytes and renal epithelial cells. This can be achieved by decreasing cholesterol synthesis via inhibition of the expression of sterol-regulatory element-binding protein 2 (SREBP-2) and β-Hydroxy β-methylglutaryl-CoA (HMG-CoA) [71] and promoting cholesterol efflux via increasing expression of ATP-binding cassette transporterA1 (ABCA1) [72]. On the other hand, FXR lowered hepatic and renal triglyceride accumulation and plasma triglyceride levels in insulin resistance models, such as ob/ob and KK-Ay mice [59], by reducing the expression of fatty acid synthase (FAS) and acetyl CoA carboxylase (ACC). This was achieved by inhibiting SREBP-1c and carbohydrate response element binding protein (ChREBP) [73,74,75], promoting triglycerides clearance by increasing fatty acids oxidation via PPARα/γ- carnitine palmitoyltransferase I (CPT1) axis [76,77] and reducing fatty acid uptake by reducing the expression of CD36 [78]. In adipocyte, FXR induces brown adipose tissue (BAT) whitening, presenting with large intracellular lipid droplets and extracellular collagen deposition as a result of activation of stearoyl-coenzyme A desaturase (SCD) expression via PPARγ activation [79,80,81,82] (Figure 1).

## 5. FXR and Renal Physiology

The kidney is responsible for the excretion of urine, allowing toxins, metabolic waste products and excess ion to be excreted while maintaining essential substances in the blood and regulating several body systems such as intra and extracellular volume status, acid-base status, calcium and phosphate metabolism or erythropoiesis. It also produces renin for blood-pressure regulation and expresses 1α hydroxylase for conversion of vitamin D to its active form. Each day, an adult human produces ∼1.5 L of urine despite 180 L of fluid being filtered through the glomerular basement membrane. Approximately 99% of glomerular filtrate is reabsorbed constitutively along successive segments of the nephron. In renal medullary collecting duct cells (MCDs), water transport is tightly regulated by arginine vasopressin (AVP), a circulating hormone also known as antidiuretic hormone (ADH). Dehydration causes a decrease in fluid volume and an increase in blood osmolality, and this water-restricted condition promotes the release of hypothalamic AVP into the blood. AVP stimulates the water reabsorption in kidney by increasing gene expression and apical membrane targeting of aquaporin 2 (AQP2) [83]. 

FXR is widely expressed in all segments of renal tubules, with relatively higher expression in the proximal tubules and thick ascending limbs, followed by the distal tubules, thin descending limbs and collecting ducts. FXR plays a critical role in the regulation of urine volume, and its activation increases urinary concentrating capacity, mainly by upregulating its target gene AQP2 expression in the collecting ducts [84]. Renal medulla is a unique tissue in which residing cells including MCDs are exposed to the harsh hypertonic and hypoxic environment and must survive significant increases in NaCl and urea concentrations during antidiuresis. FXR can ameliorate hypertonic apoptosis of MCDs by activating tonicity response enhancer-binding protein (TonEBP) and its target gene, crystallin zeta (CRYZ) [85,86]. It is reported that deletion of transcription factor hepatocyte nuclear factor-1β (HNF-1β) in the mouse renal collecting ducts (CDs) induced polyuria and polydipsia. Chromatin immunoprecipitation and sequencing experiments revealed HNF-1β binding to FXR gene promoter. This study revealed a novel transcriptional regulator of FXR in maintaining urine concentration [87]. These studies have shown that FXR plays a critical role in urine concentration by increasing water reabsorption and promoting the survival of MCDs in a dehydrated state. In addition, FXR may be a potential target for the treatment of diseases such as hepatorenal syndrome and cholestasis, which have high circulating bile acid levels. 

Several studies revealed that bile acid transporters were expressed in the kidney. Bile acids are reabsorbed by ASBT in the apical membrane of proximal tubular cells and excreted by MRP2 and MRP4 in the apical membrane and OSTα/OSTβ in the basolateral membrane, while MRP3 is not expressed to a significant level in mouse kidney [88]. The activation of renal FXR promotes bile acid excretion from urine by increasing the expression of MRP4 in the apical membranes [89] and OSTα/OSTβ in the basolateral membrane [90]. FXR have little effect on renal MRP2 and ASBT [88,90,91]. These observations suggested that in the kidney, FXR may contribute to urinary bile acid elimination, especially during cholestasis.

The kidney plays a central role in blood-pressure regulation by causing vasoconstriction and controlling circulating blood volume. Zhu et al. reported that in the mouse renal collecting duct cells mIMCD-K2, FXR agonists promoted NO production by enhancing the expression of neuronal nitric oxide synthase (nNOS) and inducible nitric oxide synthase (iNOS), whereas this effect was diminished by FXR knockdown [92]. In addition, they also investigated the anti-hypertensive effect of renal FXR on a hypertension mouse model induced by 20% fructose in drinking water with 4% sodium chloride in diet (HFS) for 8 weeks. They found that renal FXR overexpression significantly attenuated hypertension and increased renal NO levels. Moreover, chenodeoxycholic acid (CDCA), a natural ligand of FXR, attenuated elevated blood pressure in spontaneously hypertensive rats (SHR), partially through stimulating endothelial nitric oxide synthase (eNOS) expression [93,94]. These findings support a potential role of FXR in regulation of blood pressure and development of treatment for hypertension (Figure 2).

## 6. FXR and Kidney Diseases

### 6.1. Acute Kidney Injury

Acute kidney injury is a syndrome characterized by rapid loss of renal excretory function, usually diagnosed by the accumulation of end products of nitrogen metabolism (urea and creatinine) or decreased urine output. Classification of AKI includes pre-renal, intrinsic and post-renal kidney injury. Current models of AKI can be induced by ischemia–reperfusion (pre-renal acute kidney injury), injection of drugs, toxins or endogenous toxins such as cisplatin and triptolide (intrinsic acute kidney injury) and ureteral obstruction (post-renal acute kidney injury) [95]. FXR plays an important role in various types of acute kidney injury.

In renal ischemia–reperfusion (I/R) injury mice models, OCA and GW4064 can improve renal structural and function damage, decrease inflammation and apoptosis and attenuate renal oxidative and ER stress. In primary cultured mouse renal proximal tubular cells, FXR activation resulted in markedly decreased oxidative stress, mitochondrial damage and ER stress in response to hypoxia [96]. OCA also prevented the subsequent progression of AKI to chronic kidney disease (CKD) by ameliorating glomerulosclerosis and interstitial fibrosis [96,97]. Moreover, alisol B 23-acetate (ABA), a major active triterpenoid extracted from alismatis rhizome, can activate FXR and improve renal function, reducing renal tubular apoptosis by ameliorating oxidative stress and suppressing inflammatory factor expression [98]. Kim et al. found that FXR deficiency mice exhibited increased renal apoptosis and autophagy compared to wildtype mice. Meanwhile, in HK2 cell (human renal proximal tubular epithelial cells), treatment with GW4064 and OCA inhibited hypoxia-induced autophagy [97]. Controversially, it is reported that FXR-deficient mice have less kidney dysfunction, with significantly lower levels of Cr and BUN compared with wildtype mice after I/R injury. FXR deficiency attenuated I/R (hypoxia for 24h followed by 6 h of reoxygenation)-induced apoptosis in HK2 cells by increasing PI3k/Akt-mediated Bad phosphorylation [99]. Based on these studies, the effects of FXR on I/R induced renal injury and cell apoptosis are inconclusive, and more evidence is needed. AKI induced by I/R involves multiple mechanisms, including renal tubular epithelial cells and immune/inflammatory cells. Cell-specific knockout would help elucidate the role of FXR in renal I/R injury. 

In cisplatin-induced AKI, knockout of the FXR gene aggravated renal injury, the mechanism of which might be related to inhibiting autophagy and promoting apoptosis [100]. Meanwhile, OCA protected against cisplatin-induced inflammation, apoptosis and fibrosis in the kidney by regulating small heterodimer partner (SHP) [101]. In addition to the important roles of apoptosis and autophagy in acute kidney injury, recent studies have shown that ferroptosis is also involved in its occurrence. In cisplatin-induced AKI, the expression of FXR and GPX4 (the central regulator of ferroptosis) decreased while lipid peroxidation increased. FXR agonist GW4064 protected against cisplatin-induced acute kidney injury by regulating the transcription of ferroptosis-related genes [102]. Current studies showed that FXR agonist GW4064 reduced renal lipid deposition by increasing fatty acid oxidation via PPARγ, thereby ameliorating cisplatin-induced acute kidney injury [77]. Other studies reported that OCA pretreatment alleviated LPS-induced renal dysfunction and pathological damage by repressing inflammation and oxidation [103]. Dioscin, a natural saponin derived from various herbs, acted as a novel and potent FXR agonist, which suppressed inflammation and oxidative stress against doxorubicin-induced nephrotoxicity [104]. Therefore, FXR offers a promising therapeutic target for drug-induced AKI.

### 6.2. Chronic Kidney Disease 

Chronic kidney disease is defined as kidney damage or glomerular filtration rate (GFR) < 60 mL/min/1.73 m^2^ for 3 months or more, irrespective of cause. A progressive disease with no cure and high morbidity and mortality, chronic kidney disease is common in the general adult population, especially in people with diabetes and hypertension [105]. 

Diabetic kidney disease is the leading cause of CKD worldwide [106]. It is characterized by glomerular hypertrophy, mesangial expansion, tubulo-interstitial fibrosis and inflammation, glomerulosclerosis, kidney fibrosis and podocyte loss [107]. FXR is markedly decreased in both glomeruli and tubules in human kidney with diabetic kidney disease [108]. In db/db mice with type 2 diabetes, FXR agonist GW4064 or INT-767 attenuated podocytes injury, glomerulosclerosis, mesangial expansion and tubulointerstitial fibrosis by downregulating the pro-fibrotic and pro-inflammatory genes. They also significantly attenuated abnormal lipid metabolism by reducing the gene expression related to synthesis of fatty acids such as SREBP-1, while increasing the expression of ABCA-1, a modulator for cholesterol efflux [74,109,110]. In addition, tauroursodeoxycholic acid (TUDCA) and WAY-362450 (FXR-450/XL335)—both potent agonists of FXR—alleviated glomerular and tubular damage by decreasing serum TG levels and inhibiting ER stress in db/db mice [111,112]. Moreover, in db/db mice, GW4064 ameliorated the progression of diabetic nephropathy by downregulating visfatin expression, which is a recently discovered adipocytokine that has been shown to have an important role in the pathogenesis of diabetic kidney disease [113]. OCA also improved renal injury by decreasing proteinuria, glomerulosclerosis and tubulointerstitial fibrosis in mice with type 1 diabetes [114]. FXR activation-mitigated tacrolimus induced diabetes mellitus by inhibiting gluconeogenesis and promoting glucose uptake of renal cortex proximal tubule epithelial cells in a PGC1α/FOXO1-dependent manner [115]. In summary, FXR agonists may prevent diabetic kidney disease by improving renal lipid accumulation, glucose metabolism disorders, inflammation and fibrosis.

Obesity-related renal disease is associated with structural and functional changes in various kinds of renal cells, including mesangial cells, podocytes and proximal tubular cells. Obesity has become a worldwide epidemic and is an independent risk factor for development and progression of CKD. Renal lipid metabolism disorder is the main feature of obesity-related renal disease. In mice with diet-induced obesity, INT-767 and OCA prevented abnormal renal lipid metabolism, mitochondrial dysfunction, oxidative stress, inflammation and fibrosis [74,116]. Trans chalcone is a flavonoids precursor that activated FXR, significantly inhibited HFD-induced decreased insulin sensitivity and increased expression levels of SREBP-1c, FAS, MAD-3 and NGAL, thereby acting as a renal protective agent [75]. In addition, the offspring of obese mothers are more likely to develop impaired glucose tolerance and CKD, with associated downregulation of renal FXR expression and upregulation of monocyte chemoattractant protein-1 (MCP-1) and transforming growth factor-β1 (TGF-β1) [117]. In fructose-fed Wistar rats, high-fructose feeding might cause lipid nephrotoxicity through downregulated FXR, but FXR agonist CDCA modulated renal lipid metabolism, decreased proteinuria, improves renal fibrosis, inflammation and oxidative stress [118]. In uninephrectomized obese mice, OCA attenuated renal injury, renal lipid accumulation, apoptosis and lipid peroxidation [119]. Thus, activation of FXR may be a novel strategy for treatment of obesity-associated renal disease.

Obstructive kidney disease is typically simulated via ureteral obstruction (UUO) in animal models, causing renal fibrosis, a common pathway for most chronic kidney disease to progress to end-stage renal failure. In UUO mice, FXR agonist protected against renal fibrosis and downregulated Smad3 expression, which is a critical signaling protein in renal fibrosis [120]. In addition, the activation of FXR suppressed renal fibrosis by inhibiting the phosphorylation of Tyr416-Src (proto-oncogene tyrosine-protein kinase) and increasing Ser127 phosphorylation and cytosolic accumulation of yes-associated protein (YAP) [121]. Another FXR agonist, EDP-305, also reduced interstitial renal fibrosis by increasing inhibitory YAP phosphorylation in UUO mice [122]. Therefore, targeting FXR protected against renal fibrosis in a ureteral obstruction animal model.

Hepatorenal syndrome (HRS) is the most significant hepatorenal disorder occurring in patients with advanced cirrhosis. In a rat model of HRS induced by bile duct ligation for 6 weeks, OCA treatment significantly normalized portal hypertension, glomerular filtration rate, urine output, renal blood flow, decreased ascites, renal vascular resistance, serum creatinine and the release of renal tubular damage markers, including urinary neutrophil gelatinase-associated lipocalin (uNGAL) and kidney injury moleculae-1 (uKim-1) through the inhibition of renal 8-iso-PGF2α production and the downregulation of the COX-TXA2 pathway [123]. NAFLD and NASH are also associated with an increased risk of chronic kidney disease (CKD), while Vonafexor, an FXR agonist being developed for patients with NASH, can not only improve liver fat deposition, but can also increase glomerular filtration rate, thereby improving renal function [124].

Renal cell carcinoma (RCC) is the most common type of kidney cancer [125]. Some studies have found that FXR plays different roles in normal cells and tumor cells. In HK-2 cells, FXR stimulated cell differentiation by decreasing Oct3/4 level [126,127,128,129], which is a cell differentiation marker, but had no effect on the differentiation of renal adenocarcinoma cells. Controversially, FXR also stimulated the cell growth of renal adenocarcinoma cells by downregulating cyclin-dependent kinase (CDK) inhibitors p16/INK4a and p21/Cip1, but had no effect in the normal renal cell-derived cell line [130]. Recently, a cohort of adult clear cell renal cell carcinoma (ccRCC) patients revealed that FXR activation inhibits the progression of ccRCC [131]. The role of FXR in renal neoplasia needs further study (Figure 3).

## 7. Conclusions and Outlook

FXR plays crucial roles in regulating bile acid metabolism, cholesterol homeostasis, glucose and lipid metabolism. Emerging evidence demonstrates that FXR agonists are functional for metabolic syndrome and are considered potential therapeutic agents. FXR is widely expressed in the kidneys. Under physiological conditions, it plays important roles in urine concentration by promoting water reabsorption and cell survival under hyperosmotic stress. Under pathological conditions, FXR mitigates apoptosis, autophagy and ferroptosis of tubular epithelial cells, attenuates abnormal glucose and lipid metabolism and improves inflammation, oxidative stress, ER stress and fibrosis in various kidney diseases. Thus, targeting FXR has great application prospects in the treatment of various kidney diseases. However, little is known about the mechanisms by which the different FXR isoforms regulate specific genes and how the expression of these target genes affects the occurrence and development of diseases. Moreover, the kidney is a complicated organ made up of multiple cell types; the cell-specific functions of FXR also need to be elucidated in the future.

## Figures and Tables

**Figure 1 ijms-24-02408-f001:**
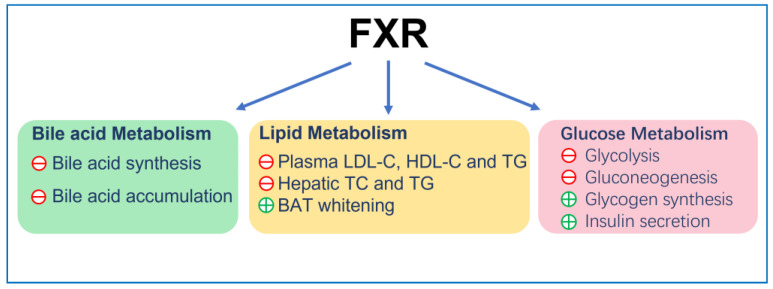
The general function of FXR. ㊉ indicates stimulation; ㊀ indicates inhibition.

**Figure 2 ijms-24-02408-f002:**
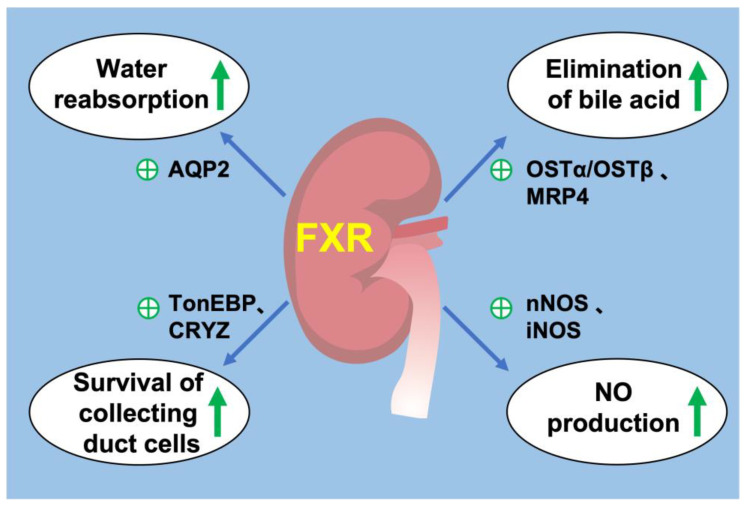
FXR and renal physiology. (1) FXR activation promotes water reabsorption in renal collecting duct by upregulating AQP2 expression. (2) FXR promotes the elimination of bile acids from the kidney by promoting MRP4 in the apical membrane and OSTα/OSTβ in the basolateral membrane. (3) FXR increases the expression of TonEBP and CRYZ to promote the survival of renal collecting duct under hyperosmotic stress. (4) FXR promotes the production of NO by increasing the expression of nNOS and iNOS in the kidney. ㊉ indicates stimulation. green arrow indicates upregulation.

**Figure 3 ijms-24-02408-f003:**
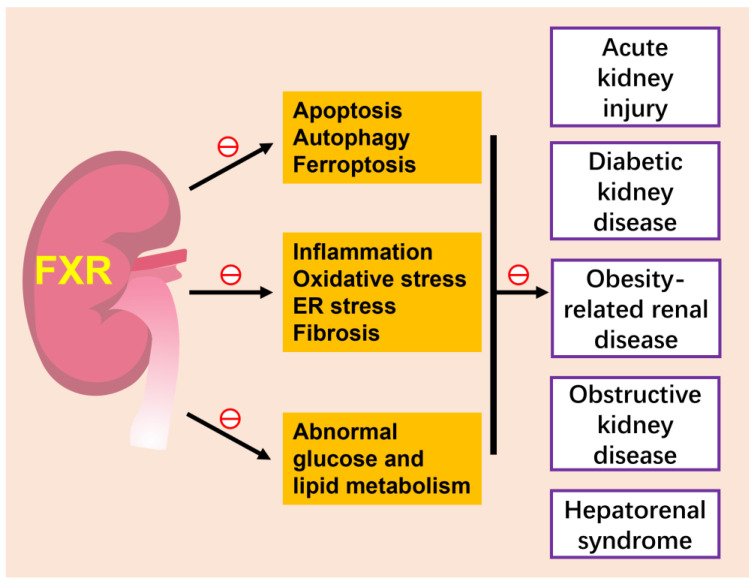
FXR and kidney diseases. The activation of FXR attenuates various kidney diseases, such as acute kidney injury, diabetic kidney disease, obesity-related kidney disease and hepatorenal syndrome, by inhibiting several cell death modes of renal tubular epithelium, including apoptotic, autophagy and ferroptosis, ameliorating inflammation, oxidative stress and fibrosis and improving abnormal glucose and lipid metabolism. The role of FXR in renal neoplasia is slightly unclear and is not included in this figure. ㊀ indicates inhibition.

**Table 1 ijms-24-02408-t001:** FXR isoforms.

Nuclear Receptors	Official Symbol	Isoforms	Exon Count	Insertion	Organism
FXRα	*NRIH4*	FXRα1 FXRα2 FXRα3 FXRα4	11 11 9 9	+ − + −	*Rodents* and *primates*
FXRβ	*NRIH5*	Five splice variants were isolated, not named	11	/	*Rodents*

Insertion: 12-bp amino acid sequence (MYTG), including (+), lacking (−). FXRα5-8 are four novel splice variants recently found in human hepatocytes and therefore not included in the table. FXRβ is a pseudogene in primates.

**Table 2 ijms-24-02408-t002:** FXR agonist clinical trials.

Agonists	Clinical Trial Status	NCT Identifiers	Indication	References
Obeticholic acid	FDA approved	NCT02308111	PBC, PSC, NASH	[21]
	Phase II	NCT01585025		
	Phase III	NCT02548351		
EDP-305	Phase I Phase II	NCT03748628	PBC, NASH	[22]
		NCT03394924		
MET409	Phase II	NCT04702490	NASH, T2DM	[27]
Cilofexor	Phase II Phase III	NCT02781584 NCT03890120	PBC, PSC, NASH	[24]
Tropifexor (LY2562175/LJN452)	Phase II	NCT02516605	PBC, NASH	[25]
Nidufexor	Phase II	NCT03804879	Diabetic Nephropathy	[26]
TERN-101(LMB-763)	Phase II	NCT04328077	NASH	[28]
EDP-297	Phase I	NCT04559126	NASH	[30]
PX-104	Phase II	NCT01999101	NAFLD	[32,33]
Vonafexor (EYP001)	Phase II Phase II	NCT03812029 NCT04365933	NASH Chronic Hepatitis B	[34]

PSC—Primary Sclerosing Cholangitis; NAFLD—non-alcoholic fatty liver disease.

**Table 3 ijms-24-02408-t003:** FXR antagonist clinical trials.

Antagonists	Clinical Trial Status	NCT Identifiers	Indication	References
Ursodeoxycholic acid	Phase IV Phase IV	NCT04617561 NCT04977661	NASH, PBC	[40,41]
Ivermectin	Phase III	NCT04529525	COVID-19	[42,43]
Benzamide	Phase III	NCT00534573	Clozapine-induced Hypersalivation	[44,45]

## Data Availability

Not applicable.

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
