# Peer review of "Role of FXR in Renal Physiology and Kidney Diseases"

_ijms, 2023, doi:10.3390/ijms24032408_

Round 1

Reviewer 1 Report

FXR have high importance in the control cell and physiological processes and description its role in the pathogenesis can be interesting for the readers and topic is suitable for IJMS. Nevertheless, some point be taken for the improvement of manuscript quality.  

Chapter 2 Table FXR protein, including variants

Chapter 4 regulation of FXR in the context cell, including schema, or figure should be added.

Chapter 6 Oncological diseases should be included.

Minor

Line 29, 82,144,1169, 174 and others uncorrected performing multiple citations

Line 165 and others missing space

Reviewer 2 Report

Manuscript by Guo et al is a well-written account of current knowledge of FXR in renal physiology and kidney diseases. It will be a highly cited article in future.

Minor comments include: 1) Inclusion of references in the first paragraph – especially for the statements in lines 24 and 26. 2) Inclusion of the regulation of FXR transcription activity (including cofactors and post-translational modifications) and its impact on renal physiology.
